# Intermittent Hypoxia on the Attenuation of Induced Nasal Allergy and Allergic Asthma by MAPK Signaling Pathway Downregulation in a Mice Animal Model

**DOI:** 10.3390/ijms23169235

**Published:** 2022-08-17

**Authors:** Doston Sultonov, Young Hyo Kim, Hyelim Park, Kyu-Sung Kim

**Affiliations:** 1Department of Otolaryngology Head & Neck Surgery, Inha University Hospital, Incheon 22332, Korea; 2Inha Research Institute for Aerospace Medicine, Inha University College of Medicine, Incheon 22332, Korea; 3Kimyounghyo ENT Clinic, 161 Shin-song-ro, Yeonsu-gu, Incheon 22002, Korea

**Keywords:** MAPK, p38, ERK, JNK, signaling, intermittent hypoxia, allergic response

## Abstract

Intermittent hypoxia (IH) has been an issue of considerable research in recent years and triggers a bewildering array of both detrimental and beneficial effects in several physiological systems. However, the mechanisms leading to the effect are not yet clear. Consequently, we investigated the effects of IH on allergen-induced allergic asthma via the mitogen-activated protein kinase (MAPK) signaling pathway. Forty BALB/c mice were dived into four groups. We evaluated the influence of IH on the cell signaling system of the airway during the allergen-induced challenge in an animal model, especially through the MAPK (mitogen-activated protein kinase) pathway. The protein concentrations of *p*-ERK/ERK, *p*-JNK/JNK, *p*-p38/p38, and pMEK/MEK were significantly reduced in the allergen-induced+IH group, compared to the allergen-induced group (*p*-value < 0.05 as considered statistically significant). The number of eosinophils, neutrophils, macrophages, and lymphocytes in the bronchoalveolar lavage fluid and Dp (Dermatophagoides pteronyssinus)-specific IgG2a and interleukins 4, 5, 13, and 17 were significantly reduced in the Dp+IH group, compared to the Dp group. These findings suggest that the MAPK pathway might be associated with the beneficial effect of IH on the attenuation of allergic response in an allergen-induced mouse model.

## 1. Introduction

*Allergic Asthma* is considered one of the most common chronic diseases in the world. More than 350 million people suffer from this multifactorial disease [1]. This disease usually presents with wheezing, cough, and dyspnea. In the pathophysiology of asthma, genetic and environmental triggers play a pivotal role, which modulate the activation and regulation of the Th2 immune response. Bronchial inflammation, smooth muscle spasm, and mucus production are triggered by IL-4, IL-5, and IL-13, which are released by the Th2 cells. Because of its complicated nature, the treatment of asthma is a very difficult process. Recently, there has been a focus on the effect of hypoxia condition on the management of allergic asthma.

The hypoxic reaction mediated by hypoxia-inducible factor-1 (HIF-1) stimulates cell survival during inflammation [2]. Recent studies outlined that the nonspecific HIF inhibitor 2-methoxyestradiol reduced the allergic pulmonary inflammatory and airway remodeling responses to ovalbumin in a mouse model [3].

The impact of the hypoxia in asthma is complicated and context-dependent, with a mild hypoxic response restricted to airway epithelium acting as a protective mechanism and exaggerated diffuse hypoxic response being a highly proinflammatory and proasthmatic reaction [2].

Our research provides compelling evidence for a role of intermittent hypoxia in the development of allergic lung inflammation in the mouse model. Importantly, we demonstrate that IH exposure to an allergen-induced mouse model leads to a decreasing expression of immune cells, such as bronchoalveolar lavage (BAL) cells, eosinophils, neutrophils, macrophages, lymphocytes, IL-4, IL-5, IL-13, and IL-17.

*Intermittent Hypoxia (IH)* has been a subject of considerable research over the past few decades. However, IH and its biological effects have not yet been fully understood. Whereas some research claims that IH contributes to pathologic medical conditions [4,5,6], other research focuses on its curative properties [4,5,7,8]. Repeated episodes of hypoxia interspersed with normoxic intervals are the key components of IH [5].

Several adaptive reactions to hypoxia involve the regulation of specific genes, and these responses differ depending on whether hypoxia is acute, chronic, or intermittent [9]. Transcription of those genes is regulated by intercellular signaling pathways, which regulate gene transcription via phosphorylation of transcription factors, such as the mitogen-activated protein kinase (MAPK) pathway [9].

*The MAPK pathway* is an essential way to switch from extracellular signals to intracellular responses. As a result of genetic and epigenetic changes, signaling cascades are altered in many disorders, including in cancer and allergies. Several studies examined both the pathogenetic and homeostatic conduct of MAPK signaling. However, there is still much to be analyzed in terms of regulation and action models in both preclinical and clinical research. The current paper discusses new insights into MAPK as a complicated cell signaling pathway with roles in maintaining a normal cell channel, response to an allergy, and activation of compensatory pathways [10]. The components of cellular signaling interact in a switch-like manner, with the interaction of two proteins causing either direct or indirect activation or inhibition of the subsequent factor [11]. The outline of the MAPK signaling cascade consists of the interaction of one or more growth factor proteins (GFs) with their specific growth factor receptors [12]. Three protein kinases (RAF, MEK, and ERK) and a little G protein (RAS) consist of the pathway’s general structure [13]. The activation cascade is in the following order: MAPKKK (mitogen-activated protein kinase kinase kinases, represented by RAF and its variants), followed by MAPK kinase (MEK1/2/3/4/5/6/7), and ultimately by the MAPK. There are three main classical MAPKs with different isoforms, which are ERKs (extracellular signal-regulated kinase 1/2), JNKs (c-Jun N-terminal kinases with JNK1, JNK2, and JNK3 isoforms), and p38 (p38α, p38β, p38γ, and p38δ) [10]. ERK is involved in a wide range of processes, for instance, survival, proliferation, and differentiation, all of which are based on the phosphorylated targets of ERK. This can activate transcription factors within the nuclease [10]. As a potential therapeutic intervention, the nuclear translocation of the MAPK signaling components is also considered as an essential regulatory system of key cellular processes [11].

The authors hypothesized the changes in the MAPK cascade might be associated with IH. Therefore, we aimed to find out the effect of IH in the allergen-induced cells, thereby revealing the molecular changes in the MAPK cell signaling pathway in an IH-induced mouse model.

## 2. Results

### 2.1. IH Reduced Airway Inflammation in Mice with Dp-Induced Allergic Asthma

Recruitment of inflammatory cells near the peribronchiolar in the Dp (*Dermatophagoides pteronyssinus*)-induced group was significantly higher than that in the CTL (control) group. On the other hand, in the Dp+IH group, the recruitment of these inflammatory cells was markedly reduced by hypoxia. To examine the effect of hypoxia on mucin secretion, periodic acid-Schiff (PAS) staining was performed. The number of goblet cells in the airway increased in the Dp group but decreased in the Dp+IH group (Figure 1).

Compared to the CTL group, the Dp group showed a significant increase in the number of inflammatory cells in the bronchoalveolar lavage fluid (BALF). In contrast, the number of inflammatory cells in the BALF was significantly reduced in the Dp+IH group. In particular, the number of eosinophils, neutrophils, macrophage, and lymphocytes decreased significantly in the Dp+IH group compared to that in the Dp group (Figure 2).

### 2.2. Intermittent Hypoxia Inhibited Cytokine Production in Allergic-Induced Asthma Model

An enzyme-linked immunosorbent assay revealed that IgE, Dp-specific IgE, IgG1, and IgG2a were significantly increased in the serum of mice in the Dp group, but IgG2a was reduced in the Dp+IH group by exposure to IH (*p* < 0.05) (Figure 3A).

To determine the manner in which IH could influence cytokine secretion in BALF, ELISA (enzyme-linked immunosorbent assay) was performed to detect IL-4, IL-5, IL-13, and IL-17 cytokine levels. The Dp group showed a statistically significant increase in the release of IL-4, IL-5, IL-13, and IL-17 in BALF compared to that of the CTL group (*p* < 0.001 for each). On the other hand, the titer of these cytokines decreased significantly with exposure to IH (*p* < 0.001 for each) (Figure 3B).

### 2.3. IH Inactivated the MAPK Pathway in Dp-Induced Asthma

As previous literature on the topic has stated, MAP kinases are regulated by phosphorylation cascades [14]. We investigated both phosphorylation and total types of ERK, p38, JNK, and MEK proteins. Proteins were extracted both in normoxia conditions and 3 days after the IH condition from the two groups of cells, which are allergen (Dp)-induced and the CTL group, respectively. To evaluate the expression and activation of MAP kinases, Western blot was carried out, and we analyzed the ratio of phosphorylated and total enzyme indicators. The results show that the Dp allergen caused a high expression of all ERK (331 ± 32%, *p* < 0.0001), JNK (211 ± 16%, *p* < 0.0001), p38 (176 ± 14%, *p* < 0.0001), and MEK (135 ± 2%, *p* = 0.0023) proteins compared to the controls (118 ± 18%, 109 ± 15%, 91 ± 8%, 80 ± 17%, respectively). To investigate the IH effects on the MAP kinases, after 3 days of IH exposure, ERK, JNK, p38, and MEK levels were re-examined. The ratio of all investigated phosphorylated and total enzymes after 3 days of IH exposure showed a further significant decrease in the allergen-induced group (Figure 4B–E). Three days of the IH cycle affected every protein differently, for instance, the phosphorylation of the ERK and p38 MAPK enzymes were increased in the CTL + IH group in comparison with the normoxia group (from 118 ± 18% to 197 ± 22% and from 91 ± 8% to 114 ± 6%, respectively). While JNK (109 ± 15%) and MEK (80 ± 17%) expression and phosphorylation showed a marginal decrease (to 107 ± 10% and 68 ± 16%, respectively) after IH exposure in the control groups. However, in the allergen-induced group, IH effects the downregulation of all MAPKs. IH exposure affected the phosphorylation and activation of ERK, JNK, and p38 kinases the same as by downregulation in the Dp group compared to controls (Figure 4B–D), whereas the *p*-MEK/MEK was slightly higher (104 ± 5%, *p* = 0.027) after 3 days of IH in comparison with the CTL + IH group (68 ± 16%) (Figure 4E). By contrast, despite the expression of *p*-MEK/MEK showed a slight decrease in the Dp+IH group (104 ± 5%, *p* = 0.027) than in the Dp group (135 ± 2%, *p* = 0.0023), it was still higher in comparison with the controls (Figure 4E). These findings indicate that IH exposure causes a decrease in allergic response (to Dp) by downregulation of the phosphorylation of MAPK pathway enzymes, such as ERK, JNK, p38, and MEK.

In the Dp group, we saw a significant decrease in all groups after 3 days of IH. The expression of *p*-JNK/JNK and *p*-p38/p38 significantly decreased in the Dp group after IH exposure from 211 ± 16% and 176 ± 14% to 98 ± 7% (*p* < 0.0001) and 95 ± 6% (*p* < 0.0001), respectively (Figure 4C,D). The *p*-ERK/ERK expression showed more of a decrease in the Dp+IH group than in the Dp group from 331 ± 32% to 146 ± 6%, (*p* < 0.0001), while *p*-MEK/MEK was slightly decreased from approximately 135 ± 2% to 104 ± 5% (*p* = 0.05), which is still higher than the CTL and CTL+IH groups (80 ± 17% and 68 ± 16%) (Figure 4B,E).

## 3. Discussion

In brief, the results from the present study demonstrate that recruitment of inflammatory cells near the peribronchiolar in the Dp group was significantly higher than that in the CTL group. To examine the effect of hypoxia on mucin secretion, PAS staining was performed. The recruitment of the inflammatory cells and the number of goblet cells in the airway were markedly reduced by hypoxia in the Dp+IH group than in the Dp group. Also, the number of inflammatory cells in the BALF were significantly reduced in the Dp+IH group than in the Dp group. Furthermore, the Dp allergen caused a high expression of ERK, JNK, p38, and MEK proteins compared to the controls. IH exposure affected the phosphorylation and activation of ERK, JNK, and p38 kinases the same as by downregulation in the Dp group compared to the controls, whereas the *p*-MEK/MEK was slightly higher after 3 days of IH in comparison with the CTL+IH group (Figure 4B–E). These findings indicate that IH exposure causes a decrease in allergic response to Dp by downregulation of the phosphorylation of ERK, JNK, p38, and MEK proteins.

MAPK pathways have critical functions in almost all aspects of cell survival, differentiation, growth, and fate [15]. Therefore, any abnormalities in this pathway might lead to dysregulation of cell responses to extrinsic stress, including allergy and hypoxic conditions [15]. The ERK cascade as a member of MAPK is activated by a pair of closely related MEKs. MEK has been shown to fully activate ERK in vitro [14]. Moreover, other MAPK family kinases, such as p38, are essential for basophil activation mediated by IgE during an allergic reaction, while another kinase, JNK, plays a role in the cell response to different intracellular and extracellular stress factors [16].

The most recent studies focused on the effects of the MAPK pathway on tumor development [7,10,13,17]. Recently published research work showed that the MAPK pathway not only regulates the behavior of tumor cells, but also the behavior of surrounding normal stromal cells in the TME (tumor microenvironment) during cancer pathogenesis [7]. The specific activities of the MAPK pathway components in tumor initiation and progression vary based on the stimuli and stromal cells involved in tumor growth [7]. Furthermore, the research about the effect of hypoxia on cancer cells showed that hypoxia-inducible transcription factor α (HIF1α) affects as a transcription factor many genes involved in anaerobic metabolisms, angiogenesis, and metastasis [7]. In addition, one recent report demonstrates that in certain conditions, the MAPK signaling pathway may regulate HIF activity, in part, by affecting the levels of HIF1α [17].

Moreover, other researchers observed the relationship between IH and upper airway inflammatory diseases and proved that IH increases inflammatory cytokines, such as IL-6, IL-8, vascular endothelial growth factor (VEGF), tumor necrosis factor-α (TNF-α), and C-reactive protein (CRP) and adversely affects the upper airway mucociliary transport [6].

In line with the previously published literature on the topic, protocols described IH in the following conditions: the severity of hypoxia within episodes ranges from 2% to 16% inspired oxygen; the duration of hypoxic episodes ranges from 15–30 s to 12 h; the quantity of cycles per day ranges from 3 to 2400; the cumulative IH protocol duration ranges from no more than 1 h to between 2 and 90 days, and the most long-lasting protocols involve IH on consecutive days, although some use alternating days [5].

Clinically, they suggest that local IH can influence inflammatory diseases, such as rhinitis, chronic sinusitis, or even viral infections in the upper airway [6]. Other authors demonstrated that chronic intermittent hypoxia (CIH) led to bronchial hyperreactivity, increased airway and systemic inflammation, and thus promoted the risk of refractory asthma [18]. In addition, according to the studies, treatment with p38 MAPK inhibitor substantially decreased the inflammation scores in an asthma mice model [8]. Also, CIH decreased the glucosteroid sensitivity by activating the p38 MAPK signaling pathway. Additionally, ERK and JNK phosphorylation and activation also can be affected after stimulation with glucosteroids, although the mechanism of this regulation is not fully elucidated [19]. Other studies have shown that CIH exposure led to the activation of the p38 MAPK pathway in the nervous system and vascular endothelium [8]. From this point of view, our study also showed that IH induced ERK and p38 MAPK activation by phosphorylation, at the same time with no significant changes in JNK phosphorylation, even though the results show downregulation of MEK phosphorylation. However, IH exposure has a positive effect on the allergic response of the nasal mucosa and decreased the MAPK factors, such as ERK, JNK, p38 MAPK, and MEK.

Previous animal studies showed that CIH may induce an obstructive airflow inflammation independent of allergen exposure. This effect was found in a 14-day exposure of IH but was not found in the 7-day IH exposure, indicating that time is a major determinant for eliciting a hyperreactive response in the airways. This finding can be elucidated by the hypothesis that different durations of IH exposure provoked different patterns of response [18].

Furthermore, other authors studied the effect of CIH on the lower airway during the allergen-induced challenge and hypothesized that CIH of obstructive sleep apnea might change the airway inflammation to a monocyte/macrophage-predominant phenotype and trigger matrix remodeling, which leads to airflow obstruction and may aggravate asthma [20]. They found that chronic IH exposure in a rat model of allergen-induced airway inflammation resulted in alterations in the airway immune reaction, which led to heterogenous structural effects of collagen deposition and matrix remodeling of the airways and lung parenchyma [20].

The results from the present study suggest that the MAPK pathway may mediate airway inflammation in allergic asthma as a potential anti-inflammatory signaling. However, there was no evaluation of the accurate count of inflammatory cells using flow cytometry. The levels of T-cell activation in the pathogenesis of HDM-induced pulmonary inflammation have also not been investigated. It is necessary to confirm the anti-inflammatory effect after MAPK signaling inhibitor treatment in alveolar macrophages. These limitations should be addressed in the near future.

## 4. Materials and Methods

### 4.1. Study Subjects

Six-week-old female BALB/c mice (18–20 g) were purchased from Orient Bio (Seongnam, Korea). All mice were housed in a specific pathogen-free environment and received allergen-free water and food. They were also able to move freely without restriction and lived in 12 h day–night cycle lighting. All experimental procedures were carried out with the approval of the Institutional Animal Center and Use Committee of Inha University. (IACUC, approval number INHA 180303-541). Mice were anesthetized by intraperitoneal injection of ketamine and xylazine (90 mg/kg and 5 mg/kg, respectively). A total of 40 mice were used in the present study.

### 4.2. Induction of Allergic Asthma

The mice in the control group (*n* = 10) received intranasal administration with normal saline seven times. They lived in normoxia conditions throughout the experiment. *Dermatophagoides pteronyssinus* is the most common aeroallergen of house dust mites. For the induction of allergic asthma, mice in the Dp group (*n* = 10) were sensitized intranasally using 50 μg Dp extract (*Dermatophagoides pteronyssinus*, PROLAGEN, Korea) and resuspended in phosphate-buffered saline (PBS). Mice in the Dp+IH group (*n* = 10) received Dp administration and were, at the same time, exposed to IH for three days. Mice in the CTL+IH group (*n* = 10) were exposed to IH without Dp administration.

### 4.3. Intermittent Hypoxia Condition

IH system is a custom-designed, computer-controlled incubation system. We used a specially designed exposure chamber to generate hypoxic conditions and to provide a constant temperature, humidity, and airflow. The chamber is automatically supplied with the respective gases (oxygen, nitrogen, and carbon dioxide). In the CTL group, 1000 hpa, O_2_—21% (atmospheric oxygen concentrations 21% O_2_; 5% CO_2_; 37 °C) was supplied continuously for 3 days. In the IH group, the Dp-induced mice were exposed to both normoxia (16 h) and hypoxia (8 h). Each chamber was programmed with altered normoxic (1000 hpa; O_2_—21%) and subsequent hypoxic (300 hpa; O_2_—7%) conditions. Each IH cycle lasted 30 s of exposure with 18–20 episodes per hour. These IH cycles were repeated for the total duration of the IH period (8 h per day) for 3 days. During the study period, the mice were exposed to IH for 8 h per day, and the overall experimental protocol is summarized in Figure 5.

### 4.4. BALF Analysis

One day after the exposure of IH, the mice were sacrificed, and the trachea was immediately cannulated with an intravenous polyethylene catheter equipped with a 20-gauge needle on a 1 mL syringe. The lungs of the mice were washed twice with 0.8 mL of PBS. In over 90% of the mice, 1 mL of BALF was recovered. The BALF was centrifuged at 1800× *g* rpm; the supernatant was removed, and the pelleted cells were resuspended in PBS. BALF cells were counted with a hemocytometer in the presence of trypan blue staining. Cytospins were prepared from resuspended BALF cells by centrifuging them onto microscope slides. The slides were stained with “Diff-quick” (Thermo Electron Corporation, Pittsburg, PA, USA) and quantified for differential cell counts by counting a total of 100 cells/slide at 40 × magnification.

### 4.5. ELISA

To detect the secretion of cytokines in whole lung tissue, 500 μL of PBS was added to an equivalent amount of lung tissue and homogenized using a hand-held homogenizer. The lysates were centrifuged at 15,000× *g* for 10 min, and the supernatant were stored at −80 °C. The secretion of cytokines (IL-4, BD Biosciences, San Diego, CA; IL-5, R&D Biosystems; IL-13, BD Biosciences; IL-17, BD Biosciences) in lung supernatant and total IgE (BD Biosciences) in serum were measured according to manufacturer’s protocol.

To evaluate Dp-specific IgE, IgG1, and IgG2a in the serum, 96-well plates were coated with Dp (10 μg/well) overnight at 4 °C. On the next day, the wells were washed three times, and blocking solution was added, and the plates were incubated at 37 °C for one hour. Mice serum samples diluted with diluent solution were added to the wells and incubated at 37 °C for two hours. The plates were then washed four times with PBS/Tween 20 and biotinylated anti-IgE (BD Biosciences), biotinylated anti-IgG1 (BD Biosciences), or biotinylated anti-IgG2a (BD Biosciences) were added to the wells and incubated at 37 °C for 1 h. After washing, 100 μL of diluted horseradish peroxidase-conjugated antibody was added to each well, and the plates were incubated at 37 °C for 30 min. The plates were then washed four times, and 100 μL of tetramethylbenzidine substrate solution was added to each well. The plates were kept in the dark at room temperature for fluorescence development, and the enzymatic reaction was stopped by adding 100 μL of stop solution. Absorbance was measured at 450 nm using an ELISA plate reader.

### 4.6. Histopathology

To evaluate pathological changes in lung parenchyma, the left lungs of mice were fixed in 10% neutral buffered formalin and embedded in paraffin. Sections (3 μm) were cut and subjected to hematoxylin and eosin staining. Sections were also subjected to PAS staining (Diagnostic Biosystems, Pleasanton, CA, USA) to evaluate the degree of goblet cell proliferation in the bronchial epithelium. Pulmonary fibrosis was assessed using Masson’s trichrome staining. A light microscope (Olympus BX43, Olympus Corporation, Tokyo, Japan) was used to perform pathological analysis on the inflammatory cells and epithelial cells in the airway of each lung lobe.

### 4.7. Protein Extraction and Western Blot

Samples were obtained from the mice’s lung tissues. Briefly, whole-cell protein extraction was run on a SDS gel, transferred to a polyvinylidene difluoride membrane, and incubated with primary antibodies overnight at 4 °C and then with secondary antibodies for 1 h at room temperature. The primary antibodies used were monoclonal or polyclonal antibodies against *p*-ERK (Cell Signaling Technologies), ERK (Cell Signaling Technologies), *p*-JNK (Cell Signaling Technologies), JNK (Cell Signaling Technologies), *p*-p38 (Cell Signaling Technologies), p38 (Cell Signaling Technologies), *p*-MEK (Cell Signaling Technologies), MEK (Cell Signaling Technologies), and β-actin (1:1000; Santa Cruz Biotechnology). The signal was developed with horseradish peroxidase-conjugated IgG using a chemiluminescent kit (Milipore, Bedford, MA, USA).

### 4.8. Statistical Analysis

Data presented in the bar graphs represent the mean values of at least four independent data points as indicated in the respective figure legends. The error bars represent the standard deviation. Graphing and statistical analyses were performed with Prism software (GraphPad Software, La Jolla, CA, USA). One-way ANOVA tests plus Bonferroni multiple comparison tests were used for single-level multiple group analysis. *p*-values < 0.05 were considered as statistically significant.

## 5. Conclusions

The impact of IH in an allergen-induced mice model was tested with a downregulated phosphorylation of ERK, JNK, p38, and MEK intracellularly. Allergen (Dp) exposure and an elevation in all MAP kinases were shown, which were all downregulated after 3 days of IH exposure and showed noticeable improvement. Taking together, our study results indicate that IH in the Dp-induced mouse model showed some inactivation in the MAPK pathway, inhibited cytokine production in BALF, and reduced airway inflammation.

## Figures and Tables

**Figure 1 ijms-23-09235-f001:**
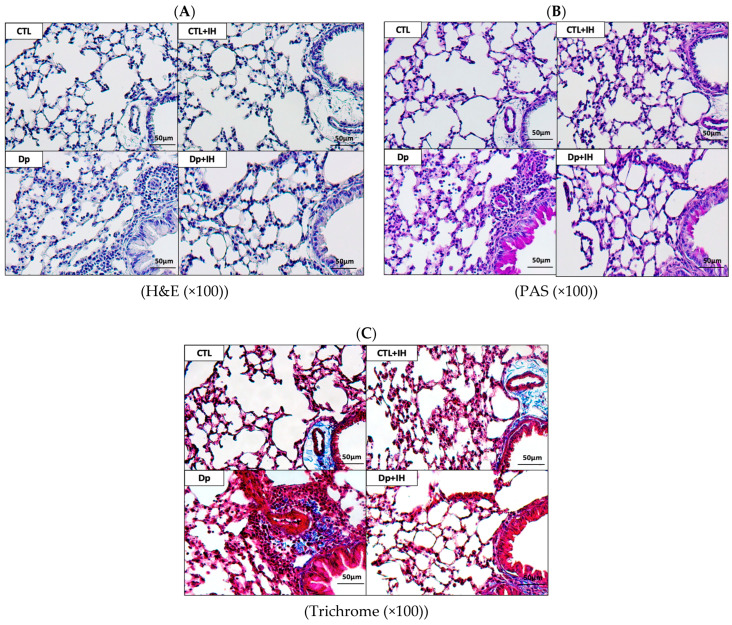
Comparison of histological changes in the lung after IH exposure. (**A**) Representative pictures of hematoxylin and eosin staining (100×) and (**B**) PAS staining (100×) (**C**) Masson’s trichrome staining (100×) of lung sections from mice after the final administration of Dp. Scale bars, 50 μm. Sections are representative of seven mice. CTL: control, saline challenge under normoxia conditions, CTL+IH: IH exposure without Dp administration Dp: Dp administration for allergic asthma induction under normoxia conditions; Dp+IH: Dp administration for allergic asthma induction and IH exposure.

**Figure 2 ijms-23-09235-f002:**
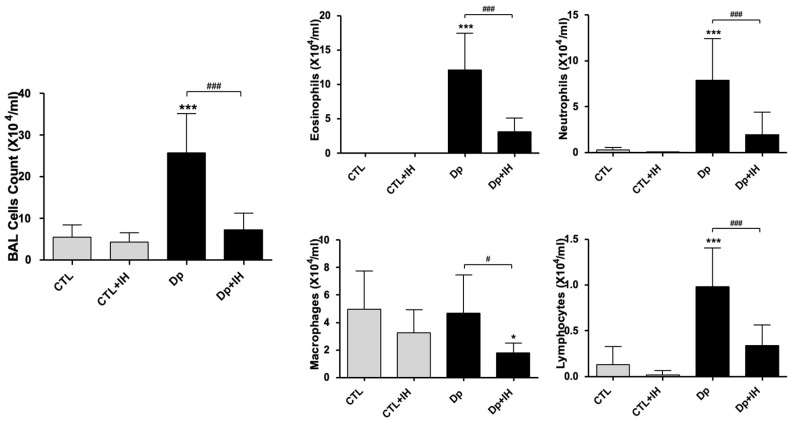
The number of inflammatory cells in the BALF. The number of eosinophils, neutrophils, macrophages, and lymphocytes. Data shown are the mean ± standard deviation (*n* = 7). * *p* < 0.05 vs. CTL, *** *p* < 0.001 vs. CTL, ### *p* < 0.001, # *p* < 0.05.

**Figure 3 ijms-23-09235-f003:**
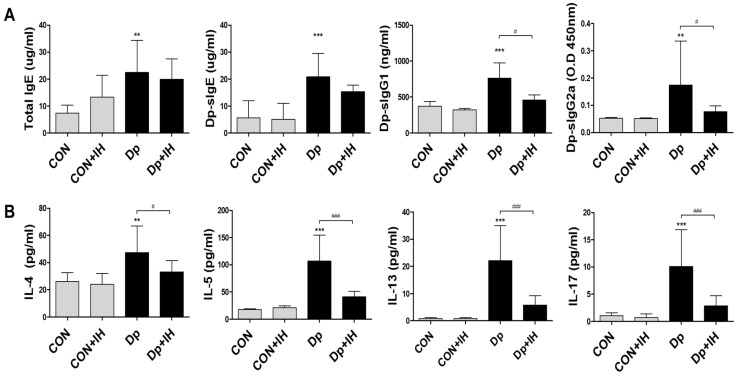
The level of Dp-specific immunoglobulin in serum and cytokine levels in BALF. (**A**) Serum IgE, Dp-specific IgE, IgG1 (at serum dilution of 1:1000), and IgG2a (at serum dilution of 1:10) were evaluated after the final administration of Dp. (**B**) Levels of IL-4, IL-5, and IL-13 were measured in the BALF by ELISA. Data shown are the mean ± standard deviation (*n* = 7). ** *p* < 0.01 vs. CTL, *** *p* < 0.001 vs. CTL, # *p* < 0.05, ### *p* < 0.001.

**Figure 4 ijms-23-09235-f004:**
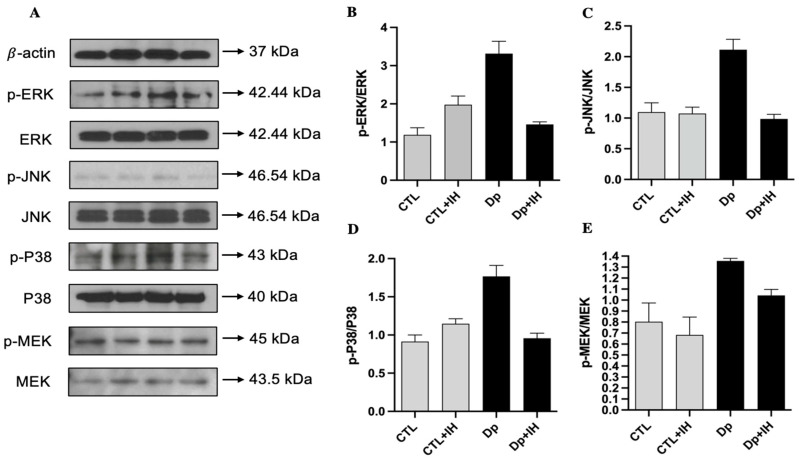
Effects of intermittent hypoxia (IH) exposure to the MAPK pathway on the allergen-induced (Dp) mice model. (**A**) Western blotting for MAPK signaling key factors ERK, JNK, p38, and MEK and their phosphorylated forms before and after 3 days of IH exposure and their molecular weight (kDa). (**B**–**E**) Relative expression of phosphorylated and total ERK (**B**), JNK (**C**), p38 (**D**), and MEK (**E**). Data in 2 groups: control (CTL) and Dp groups were compared before and after 3 days of IH exposure. Data shown are the mean ± standard deviation (*n* = 5).

**Figure 5 ijms-23-09235-f005:**
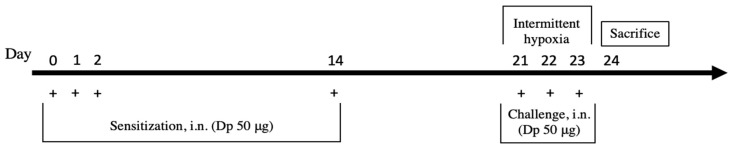
Diagram of intermittent hypoxia exposure. The mice in the control group received intranasal administration with normal saline seven times. They lived in normoxia conditions throughout the experiment. For the induction of allergic asthma, mice in the Dp group were sensitized intranasally using 50 μg Dp extract, resuspended in PBS. Mice in the Dp+IH group received Dp administration and were, at the same time, exposed to IH for three days. Mice in the CTL+IH group were exposed to IH without Dp administration.

## Data Availability

Not applicable.

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
