# Peer review of "Intermittent Hypoxia on the Attenuation of Induced Nasal Allergy and Allergic Asthma by MAPK Signaling Pathway Downregulation in a Mice Animal Model"

_ijms, 2022, doi:10.3390/ijms23169235_

Round 1
Reviewer 1 Report
The article entitled "Intermittent Hypoxia on the Attenuation of Induced Nasal Allergy and Allergic Asthma by MAPK Signaling Pathway Downregulation in a Mice Animal Model" is certainly very interesting and of scientific relevance. The authors base their morphological results exclusively on H/E and PAS reaction. This appears to be a limited result, if taken as the only morphological result. The authors should evaluate the morphological alterations using also other stains, such as Alcian Blue/ Pas to better characterize goblet cells and also highlight any quantitative and qualitative changes of specific cells, such as presence of immune cells, typical of inflammation . I suggest the authors to use different stains, including trichrome like Mallory and Masson, or histochemical stains like AB/PAS. It could be interesting to stain the sections with GIEMSA, for a general evaluation of the immune cells. Overall, the article can be appreciated and considered for publication after a revision, with the possible addition of other histological data, to improve the quality of the manuscript.
Author Response
Dear Reviewer,
Thank you for your review.
I am going to provide a short detailing of our changes for the referees’ approval.
- All authors decided to change the authorship that we wish to include Dr. Hyelim Park as a co-corresponding author. I sent an Authorship change form to the Editor.
- In Figure 2, we also included Masson’s trichrome staining and all histological figures modified to 100x magnification so that the cells were observed.
- All abbreviations and numbers were corrected according to the referees suggests. And minor English language spell check has been provided.
All authors have read and approved the manuscript revision and agree with its submission to International Journal of Molecular Sciences.
Thank you again for your review. We really appreciate it.
Sincerely,
Doston Sultonov
Reviewer 2 Report
In this manuscript, author investigated the effects of IH on allergen-induced allergic asthma via the mitogen-activated protein kinase (MAPK) signaling pathway. Most of results were interesting to understand the role of MAPK pathway on the attenuation of allergic response in an allergen-induced mouse model. The data are comprehensive and well presented in the figures, and the experimental approaches appear sound. However, the manuscript needs minor modifications to be considered by International Journal of Molecular Sciences.
Minor comments:
1) The introduction should be presented in a logical and simple way to help the reader understand. Simple basic knowledge should be deleted.
2) In figure 2, cell image should be added.
3) Author should describe the detail information about animal experiment including origin of animals, the number of animal per each group, diet information, breeding condition, anesthesia, euthanasia observation period et al.
4) All abbreviations should be fully described when it firstly appeared. Also, this description should be not repeated in text. These words were maintained the same form in all text and figure. Ex) Dp/DP, PBS/1x PBS et al.
5) All number should be separated unit except % and oC. Also, unit should be described same pattern.
6) References should be corrected according to journal guideline.
7) Why did you select Dp extract to induce allergic asthma? You should describe a reason into Martials and methods.
8) Author should describe the detail information about “mixed gases”.
Author Response
Dear Reviewer,
Thank you for your review.
I am going to provide a short detailing of our changes for the referees’ approval.
- All authors decided to change the authorship that we wish to include Dr. Hyelim Park as a co-corresponding author. I sent an Authorship change form to the Editor.
- We made a change in the introduction for a logical and simple way to help the reader understand. And most of the simple basic knowledge excluded.
- In Figure 2, we also included Masson’s trichrome staining and all histological figures modified to 100x magnification so that the cells were observed.
- We revised the Materials and methods part, given information about animals, Dp extract and “gases”.
- All abbreviations and numbers were corrected according to the referees suggests. And minor English language spell check has been provided.
- References corrected according to the journal guideline.
All authors have read and approved the manuscript revision and agree with its submission to International Journal of Molecular Sciences.
Thank you again for your review. We really appreciate it.
Sincerely,
Doston Sultonov